# Tumor Recurrence and Graft Survival in Renal Transplant Recipients with a History of Pretransplant Malignancy: A Matched Pair Analysis

**DOI:** 10.3390/jcm10112349

**Published:** 2021-05-27

**Authors:** Felix Becker, Anne-Sophie Mehdorn, Vasilios Getsopulos, Katharina Schütte-Nütgen, Stefan Reuter, Barbara Suwelack, Andreas Pascher, Jens G. Brockmann, Ralf Bahde

**Affiliations:** 1Department of General, Visceral and Transplant Surgery, University Hospital Münster, 48149 Münster, Germany; vasiliosg@gmx.de (V.G.); andreas.pascher@ukmuenster.de (A.P.); Jens.Brockmann@ukmuenster.de (J.G.B.); bahder@web.de (R.B.); 2Department of General, Visceral, Transplantation, Thoracic and Pediatric Surgery, University Hospital Schleswig-Holstein, Campus Kiel, 24105 Kiel, Germany; anne-sophie.mehdorn@uksh.de; 3Division of General Internal Medicine, Nephrology and Rheumatology, Department of Internal Medicine D, University Hospital Münster, 48149 Münster, Germany; Katharina.schuette-nuetgen@ukmuenster.de (K.S.-N.); Stefan.Reuter@ukmuenster.de (S.R.); Barbara.Suwelack@ukmuenster.de (B.S.)

**Keywords:** kidney transplantation, graft survival, oncological outcome, waiting time, cancer

## Abstract

Organ scarcity demands critical decision-making regarding eligible transplant candidates and graft allocation to ensure best benefit from renal transplantation (RTx). Among the controversial relative contraindications is a history of pretransplant malignancy (PTM). While oncological outcomes of PTM-RTx recipients are well described, data on graft-specific outcome are scarce. A retrospective double case control matched pair analysis (60 months follow-up) was carried out and RTx-recipients were stratified for history of PTM. First, PTM-RTx recipients were matched according to age, sex and duration of immunosuppressive therapy. Next, PTM-RTx recipients were matched 1:1 for age, sex and cause of end-stage renal disease. Five-year patient and graft survival as well as oncological outcomes were analyzed. A total of 65 PTM-RTx recipients were identified. Post-RTx recurrence rate was 5%, while 20% developed second de novo malignancy, comparable to 14% in the control group. PTM-RTx recipients had a noticeable lower five-year death-censored as well as overall graft survival and Cox proportional hazard modeling showed a correlation between PTM and inferior graft survival. Although underlying reasons remain not fully understood, this study is the first to show inferior graft survival in PTM-RTx recipients and advocates necessity to focus on more meticulous graft monitoring in PTM recipients in addition to heightened surveillance for cancer recurrence.

## 1. Introduction

Renal transplantation (RTx) is currently the only definitive therapy for patients with end-stage renal disease (ESRD), providing the possibility to avoid livelong renal replacement therapy (RRT) [1]. Since RTx offers superior medical outcomes including improved survival and enhanced quality of life, it is considered the preferred treatment option for eligible patients [2]. However, critical risk balancing between RRT and RTx is mandatory to identify suitable transplant candidates. One of the few but well-defined contraindications for RTx is the presence of active or past malignancy, with a few exceptions such as non-melanoma skin cancer (NMSC) or small renal incidentalomas [3,4,5]. Moreover, pretransplant malignancy (PTM) has also been considered a relative contraindication until a disease-specific minimum remission time has been achieved, varying from two years for early breast, colorectal or renal cancer up to a minimum of five years for more advanced or aggressive entities [5,6,7]. This rather strict strategy is based on the knowledge that PTM is a risk factor for occurrence of post-transplant malignancies since immunosuppressive therapies are known to foster the risk of cancer development and recurrence [5]. This is supported by data from large registries that report an overall incidence of posttransplant recurrence of up to 21.5% in PTM recipients [8,9]. However, current advances and significant improvements in oncological diagnosis and especially treatment as well as refined immunosuppressive regimes have recently led to updated recommendations concerning PTM in transplant candidates in form of a consensus expert opinion statement by the American Society of Transplantation [5,7].

In an aging society, including RTx-recipients, incidence of cancer increases and several lines of evidence demonstrate an increased risk of cancer and cancer-related death in potential transplant candidates. Currently, 7.0% of all solid organ transplant (SOT) recipients in population-based studies have a PTM [10]. This number is even expected to increase with the expansion of the eligibility criteria for older patients. Further, among RTx-recipients cancer is the second most common cause of death and assumed to be the leading cause of death in this decade [11,12].

## 2. Materials and Methods

### 2.1. Study Design

A retrospective single center study with a double case control matched pair analysis and a follow-up period of 60 months after transplantation was conducted. Ethical approval from local ethics committee was obtained (Ethik-Kommission der Ärztekammer Westfalen-Lippe und Westfälischen Wilhelms-Universität, No. 2018-502-f-S). All organs were procured on behalf of Eurotransplant in different hospitals of the Eurotransplant (ET) area. Transplantations were only performed at the Department of General, Visceral and Transplantation Surgery at the University Clinics Münster, Germany. The study was conducted in accordance with the ethical principles of the Declaration of Helsinki. Donor and recipient data were extracted from Eurotransplant Network Information System (ENIS), in-house transplant data files and patient charts [13]. Only de-identified data were used for final analysis.

### 2.2. Study Population

The initial screening included all RTx recipients at the Department of General, Visceral and Transplant Surgery, University Hospital Münster, Germany between 1 January 2000 and 31 July 2012, with a five-year follow-up ending on 1 July 2017. Exclusion criteria were recipient age < 18 and recipients with a previous history of transplantation of any kind. Remaining patients were stratified based on the history of PTM. For the first analysis, all RTx recipients with a history of PTM were matched 1:1 in a case control matched pair analysis to corresponding recipients without PTM. The data was then compared in terms of incidence of recurrence and de-novo as well as secondary de-novo malignancy (de novo malignancy after transplantation in the setting of PTM) after RTx. Matching criteria were age, sex, and duration of immunosuppressive therapy. Since using time under immunosuppressive therapy as matching criteria allows inclusion of one of the most established risk factors for developing cancer following SOT, it also entails a bias when analyzing patient and graft survival times. Therefore, a second match-pair analysis was conducted. Here, the PTM-RTX cohort was again matched 1:1 to RTx-recipients without PTM to analyze patient and graft survival. Matching criteria in the second matched pair analysis were age, sex and underlying cause of ESRD.

### 2.3. Demographics

The extracted recipients variables included age, sex, body mass index (BMI), cold and warm ischemia time, underlying cause of ESRD (hypertension, diabetes, polycystic kidney disease, obstructive nephropathie, glomerulonephritis, focal segmental glomerulosclerosis (FSGS), interstitial nephritis, vasculitis, other (acute kidney injury, bilateral nephrectomy) and unknown), dialysis vintage, immunosuppressive therapy (duration and protocol), Human Leukocyte Antigen (HLA) mismatch (A, B, DR and total), history of hypertension, diabetes or coronary artery disease (CAD). For the PTM-RTx cohort, the following additional disease specific parameters were retrospectively collected by electronic record review: primary malignancy entity (non-melanoma skin cancers (NMSC), malignant melanoma, urothelial cell, gynecologic, kidney, gastrointestinal, prostate, thyroid, breast, head & neck, lung, hematologic, neuroendocrine tumors (NETs)), tumor nodus metastasis (TNM)-stage, treatment algorithm, time of diagnosis between malignancy and transplantation and oncological follow-up according to current German oncological guidelines. The donor data included age, sex, body mass index (BMI,) AB0-status, last serum creatinine pre procurement, type of graft donation (donors after brain death (DBD) or living donors (LD)) and possible allocation within the Eurotransplant Senior Program (ESP). Since the Kidney Donor Profile Index (KDPI) and Kidney Donor Risk Index (KDRI) have been recently validated for an ET cohort, both were used to assess likelihood of graft failure, metric for donor characterization and risk [14].

### 2.4. Transplantation

RTx-recipients received grafts from DBD or LD. Grafts were allocated AB0- and HLA compatibly in case of DBD. AB0-incompatible transplantations were performed only in living donations. A negative pre-transplantation cross-match was mandatory for all cases. If possible, grafts were transplanted in the right or left iliacal fossa with arterio-venous anastomoses to the external iliacal artery and vein, respectively. Ureteral anastomosis was modified according to Lich-Gregoire and stented using a double J-catheter. Double J-catheters were left in situ for six weeks and removed transurethrally. Until 2005, patients received a suprapubic catheter and a transurethral catheter for two weeks in order to achieve no pressure urinary diversion. After 2005, only transurethral catheters were placed for five days post transplantation. The usual immunosuppressive regime for AB0-compatible RTx-recipients included a triple therapy, consisting of tacrolimus, mycophenolate mofetil and steroids in reduced doses. Basiliximab or anti-thymozyte globulin were used as induction therapy. Use of cyclosporine A was abandoned in 2006. (Val)ganciclovir was used for CMV-prophylaxis depending on CMV status of donors and recipients. Cotrimoxazol was used for Pneumocystis carinii prophylaxis for 100 days following transplantation.

### 2.5. Outcome Measures

Primary outcome for the first matched pair analysis was incidence of post-transplant malignancy (de-novo, second de-novo and recurrence). Primary outcome for the second matched pair analysis was 5-year-graft and -patient survival. Secondary outcome parameters included frequencies of delayed graft function (DGF, dialysis with in the first week after transplantation), frequencies of primary nonfunction (PNF, permanent loss of allograft function starting immediately after transplantation), episodes of biopsy proven acute rejection (BPAR) within one year after RTx and 1- and 5-year creatinine and eGFR (calculating using the CKD-EPI equation).

### 2.6. Statistical Analysis

Normally distributed continuous variables are presented as mean ± standard deviation (SD) and groups were compared utilizing the student’s *t*-test. For continuous variables which are not normally distributed, median and quartiles (interquartile range, IQR, Q_0.25_–Q_0.75_) are given and a comparison between groups was performed with the Mann-Whitney U test. For categorical variables, the Fisher’s exact test was used. One and five-year patient survival, death-censored graft and overall graft survival were estimated by Kaplan-Meier methodology [15] and compared using log-rank tests; *p*-values ≤ 0.05 were considered statistically noticeable. Cox proportional hazards regression models with univariable and multivariable logistic regression analyses of matched cohorts were used to determine independent factors influencing patient, death-censored, and overall graft survival at five years [16]. Univariable analysis included PTM, recipient age, recipient sex, cold ischemia time, warm ischemia time, dialysis vintage, cause of ESRD and number of HLA-mismatches. To adjust for multiple variables, a stepwise forward variable selection procedure (including variables with *p*-value less than 0.05 in the likelihood ratio test) was performed for the final multivariable model. Results are shown as hazard ratios (HR) with 95% confidence interval (CI) and *p*-value of likelihood ratio test. All statistical analyses were performed with IBM SPSS^®^ Statistics 24 for Windows (IBM Corporation, Somers, NY, USA).

## 3. Results

Between 1 January 2000 and 31 July 2012, 1217 patients received a RTx at our center. A total of 838 RTx patients met the inclusion criteria and were included in further analysis. Of these, 65 (8.0%) patients had a history of PTM. The average age of patients at the time of cancer diagnosis was 53.5 ± 10.9 years and the majority (*n* = 36, 55.0%) of the PTM-RTx cohort was male (Table 1). Skin (ten NMSC, three maligne melanoma) and urothelial cell cancer (*n* = 10) were the two most common types of cancer, followed by gynecological malignancy (*n* = 7) (Table 1). Assuming a different and less aggressive tumor biology in NMSC patients, all analyses were additionally conducted after excluding the ten NMSC patients. However, no significant differences were noted (data not shown) and therefore the patients were included in the final data set. Most PTM were detected and treated in TNM-stage T1 (*n* = 17). Only one patient was diagnosed with a TNM-stage T4 prostate cancer pre transplantation. RTx was performed on average 105.6 ± 78.1 months after diagnosis of malignancy. Age at RTx was 62.5 ± 8.6 years and PTM-RTx-recipients had been on renal replacement therapy (RRT) for 60.6 ± 31.7 months (min, max: 6, 468 months). Majority of malignancy had been diagnosed pre RRT (44 before (67.7%); 21 during (32.3%) RRT). Patients suffering from gynecological cancers had the longest time of RRT, whereas patients with a history of thyroid cancer had the shortest time of RRT and oncological follow-up, respectively. PTM-RTx-recipients with history of urothelial tumors had the third longest time of RRT. Yet, in these cases, it is unknown if the urothelial carcinoma itself was the reason for RRT. Triple immunosuppressive therapy was used in the majority of cases consisting of steroids (95.4%), mycophenolic acid (93.8%) and tacrolimus (80.0%). Cyclosporine A (15.4%) and mTOR-inhibitors (1.5%) were only used in selected cases. Induction therapy was given in 54 (83.1%) cases (51 times basiliximab, three times thymoglobuline). In the matched cohorts without history of PTM, induction therapy was given to 54 (83.1%) and 55 (83.3) patients, respectively.

After 12 months, 88.1% of patients were still on steroid-based therapy combined with mycophenolic acid (78.0%) and tacrolimus (62.7%) or cyclosporine A (16.9%) while five patients were switched to an mTOR-inhibitor (8.5%). There was no difference in doses applied and drug levels targets between patients with and without a previous history of malignoma. In addition, frequency of induction therapy and used drugs was also similar between PMT-RTx and RTx patients. In case of BPAR, steroid pulse therapy was applied.

To analyze the incidence of post-transplant malignancy, a matched pair analysis was conducted based on age (±5 years), sex and time under immunosuppressive drugs (±1 year). PTM-RTx and RTx recipients were matched 1:1 achieving similar results for age (PTM-RTx: 62.5 ± 8.6 years; RTx: 61.9 ± 9.1 years), sex (PTM-RTx: 55.4% male; RTx: 55.4% male) and time under immunosuppressive drugs (PTM-RTx: 1825 days; RTx: 1825 days) (Appendix A).

When donor characteristics were compared, PTM-RTx- and RTx-donor cohorts did not differ noticeably regarding demographic factors (donor age, sex, BMI and serum creatinine) and frequencies of living donation or ESP allocation. In addition, cold and warm ischemia times as well as HLA matching were comparable (Appendix A). When analyzing pre-transplant recipient factors, age, sex, BMI, cause of ESRD, numbers of patients depending on RRT and dialysis vintage were comparable in both cohorts. Comorbidities that affect graft function such as arterial hypertension, diabetes, and CAD were also similarly present (Appendix A). Analyzing the incidence of post-transplant malignancies, immunosuppressive therapy was compared and found to be comparable in both cohorts with 95.4% (PTM-RTx) and 100% (RTx) of patients receiving steroids, 93.8% (PTM-RTx) and 89.2% (RTx) MMF, respectively, and 80.0% in both cohorts tacrolimus. In addition, frequencies of induction therapy were also comparable between PTM-RTx (83.1%) and RTx (83.1%) patients.

To further investigate whether PTM-RTx-recipients were vulnerable for the development of post-transplant malignancy, matched cohorts were analyzed for frequencies of tumor recurrence, de-novo and second de-novo malignancy following RTx. It was found that 75.4% (PTM-RTx) and 86.2% (RTx) of patients showed an uneventful post-transplantation course over the study period of 60 months without developing any form of malignancy (Table 2). However, three patients in the PTM-RTx-cohort experienced recurrence of their original cancer. Recurrent tumors were urothelial, renal and NMSC, one each. Thirteen (20%) PTM-RTx-recipients developed second de-novo malignancy after RTx. Second de-novo malignancies were mainly NMSC (*n* = 9), but also lung (*n* = 2), breast (*n* = 1) and esophageal (*n* = 1) cancer. Nine (13.9%) RTx-recipients reported a history of de-novo malignancy after RTx. Leading malignancy among these were skin (*n* = 5) and one of the following: renal, lung, lymphoma and Barrett’s carcinoma. Kaplan-Meier analysis was used to display disease-free survival for patients with tumor recurrence, de-novo, and second de-novo malignancy (Figure 1). Interestingly, there was only a very slight difference in time between RTx and diagnosis of malignancy between de-novo malignancies in the RTx-cohort (884.2 ± 496.8 days), second de-novo malignancy (973.7 ± 452.4 days) days and tumor recurrence (1038.0 ± 370.9 days) in the PTM-RTx-cohort (Table 2). In addition, time from RTx to diagnosis of de-novo malignancies in the RTx-cohort was also comparable (884.2 ± 496.8 days).

Since the time-point of diagnosis in relation to RRT as well as waiting time between PTM and RTx are considered to be related to tumor recurrence and patient survival, the PTM-RTx cohort was further stratified accordingly and analyzed for patient and graft survival. The 65 patients in the PTM-RTx cohort were stratified for development of PTM before or during RRT (Appendix A) and for a defined waiting time cut off point (>/<5 years from PTM to RTx) (Appendix A). In addition, it was investigated whether time from diagnosis to RTx would influence death-censored graft survival or overall graft survival. Unadjusted Cox proportional hazard modeling revealed no association between time from diagnosis to RTx with death-censored graft survival (HR: 1.002, 95% CI: 0.996–1.008, *p*-value = 0.438)) or overall graft-survival (HR: 1.002, 95% CI: 0.997–1.007, *p*-value: 0.382). Kaplan-Meier analysis revealed comparable outcomes in 5-year patient, death-censored and overall graft survival. When analyzing the post-transplant graft-specific outcome, no differences were found regarding PNF, DGF, BPAR or one- and five-graft function (using eGFR as surrogate) or death censored graft loss (Table 2).

Disease-free survival of pretransplant malignancy (PTM) renal transplantation (RTx)-recipients and RTx-recipients. Survival rates of PTM-RTx-recipients and RTx-recipients with recurrence (red), second de-novo (blue) or de-novo (green) cancer after renal transplantation (RTx) were estimated by Kaplan-Meier methodology

Using time under immunosuppressive therapy as matching criteria allows inclusion of one of the most established risk factors for developing cancer following SOT. However, it also entails a bias when analyzing patient and graft survival times. Therefore, a second 1:1-matched pair analysis was conducted to investigate patient and graft survival. Matching criteria were age, sex and underlying end-stage renal disease (Appendix A). There were no differences between the two cohorts regarding the baseline donor characteristics (Table 3).

When analyzing recipient characteristics (Table 4), a higher BMI was found in the RTx group (PTM-RTx 25.4 ± 3.3 kg/m^2^; RTx 26.9 ± 3.8 kg/m^2^). When graft specific outcomes were compared, frequencies of PNF, DGF, BPAR as well as one and five-year eGFR were similar (Table 4). However, death-censored graft survival within the first year after RTx as well as after 5 years was lower in the PTM-RTx cohort (Table 4). Kaplan-Meier analysis was used to generate survival curves for 5-year patient (Figure 2a), death-censored graft (Figure 2b) and overall graft survival (Figure 2c) stratified for PTM-RTx and RTx. When groups were compared by log-rank test, a similar 5-year patient survival was found (PTM-RTx 78.5%; RTx 89.2%, *p* = 0.1). Yet, when 5-year death-censored and overall graft survival were estimated, noticeable differences were revealed. Patients in the PTM-RTx cohort had a lower death-censored (PTM-RTx 76.9%; RTx 93.8%, *p* = 0.006) as well as overall graft survival (PTM-RTx 63.1%; RTx 86.2%, *p* = 0.003) five years after RTx. The leading cause of death-censored graft loss in both groups was rejection, with a comparable median time from RTx to event (PTM-RTx: 534.1 ± 169.9 days; RTx 394.8 ± 184.3 days).

Next, Cox proportional hazard regression models with univariable and multivariable logistic regression analysis were used to further investigate the influence of PTM on patient and graft survival. Unadjusted Cox proportional hazard modeling showed that PTM-RTx patients had a 4.198 (1.392–12.657 95% CI, Table 5) hazard of death-censored graft loss and a 2.997 (1.393–6.541 95% CI, Table 5) hazard of overall graft loss. Multivariable Cox regression models adjusted for potential confounders revealed that PTM was still associated with an inferior death-censored (HR: 4.535, 95% CI: 1.503–13.680 and *p*-value = 0.007) as well as overall graft survival (HR: 3.233, 95% CI: 1.499–6.973 and *p*-value = 0.003).

## 4. Discussion

To the best of our knowledge, this is the first study to analyze oncological, patient- and particularly graft-specific outcome in a PTM-RTx cohort using a case control matched pair analysis in a German transplant center in the Eurotransplant region. When analyzing single center data from PTM-RTx-recipients, previously demonstrated recurrence rates were confirmed, while this study for the first time provides evidence derived from granular data for a higher early (1-year) as well as late (5-year) graft loss in PTM-RTx recipients in the Eurotransplant region.

The prevalence of cancer is steadily rising in the general population as well as among RTx-recipients. Due to RTx-specific risk factors, including ESRD, RRT, viral infections, immunosuppressive therapy, as well as general risk factors (age, sex, family history, comorbidities, and environmental factors), RTx-recipients suffer from an increased risk for malignancy [17,18,19,20]. Currently, cancer is the second most common cause of death among RTx-recipients and it is assumed to overtake cardiovascular diseases as the leading cause [11]. In comparison to the general population, RTx-recipients who suffer from cancer are reported to have impaired outcomes [9,21]. However, both a higher cancer incidence and an increased cancer-specific mortality hold true for patients undergoing RRT [22,23]. Transplant physicians therefore always have to face the dilemma dealing with different life-prolonging therapies (RRT vs. RTx) [7]. It is hence a growing challenge for medical transplant professionals to develop strategies for balancing the risk of following proposed waiting time for transplant in previous cancer patients as well as managing and surveiling PTM-RTx recipients.

Most data on outcome from PTM-Tx recipients have been provided by large registry analyses. Israel Penn was first to build a transplantation registry, the Israel Penn International Transplant Tumor Registry, now Cincinnati Tumor Transplant Registry (CTTR). It reports recurrence rates of up to 21.0% and the development of secondary malignancy in approximately 33.0% of PTM-RTx-recipients, which is in line with more recent data from Acuna et al. [8,10]. Brattström et al., on the other hand, estimated a recurrence rate of 9.4%, which is closer to the presented results here [24]. The current study identified PTM-recurrence in three (4.0%) cases, reporting recurrence rates as low as demonstrated by others [25,26]. Possible explanations for the presented difference might be based on varying waiting and follow-up times. In general, era-dependent refinements in induction and maintenance of immunosuppressive therapy could also play a role. However, both induction therapy as well as maintenance immunosuppressive regimens did not differ between the matched groups, which suggests that differences in tumor recurrence or development of de novo malignancy were not attributable to the use of induction therapy. Moreover, the low recurrence rates reported here are also influenced by a comparably small cohort as well as relatively short follow-up of five years in combination with a rather strict comply with minimum waiting time to RTx in PTM recipients.

This study provides further evidence for inferior survival rates in PTM-RTx recipients by reporting worse five-year overall patient survival. Of interest, survival rates in the RTx cohort are comparable to larger cohorts, previously published from our center, in which 5-year overall survival rates between 87.1% and 92.4% were noted. (CITATION) This provides further evidence that choosing a highly selective cohort for the matched pair analysis did not introduce a bias in the analysis. In general, our survival data confirm previous reports indicating worse outcomes in terms of cancer mortality, all-cause mortality and outcome of posttransplant de novo malignancies in PTM-RTx-recipients [6,9,24]. Brattström et al. identified an increased rate of death in PTM-RTx-recipients, mainly attributable to cancer. In a propensity score matched analysis, Acuna et al. further confirmed this trend, indicating that PTM-patients had worse overall survival compared to patients without a history of PTM if they received a graft within the first five years after diagnosis of cancer. If patients were transplanted more than five years after the diagnosis of malignancy (even non-aggressive tumors), the outcome was worse [10]. This is contradictory to the results from Brattström et al. who reported a better outcome after a longer waiting time [24]. In a larger cohort with a reported waiting time of one year, Dahle et al. observed similar all-cause mortality as well as overall survival in PTM-RTx-recipients [25]. Analyzing United Network for Organ Sharing (UNOS) data, Livingston-Rosanoff et al. as well found inferior long-term outcome in PTM-RTx recipients [9]. The data presented here could not show survival differences when patients were stratified for waiting times, which is probably at least in part attributed to a potential bias. A proportion of the PTM cohort were diagnosed and treated a long time before the start of RRT, which might itself have created a possible selection and allowed for full recovery, while other studies focused on patients with cancer diagnosis under RRT and the urgent decision process of prolonging RRT to achieve mandatory disease-free waiting times.

Regarding graft specific outcomes, Livingston-Rosanoff et al. only recently used the UNOS database to analyze outcomes from RTx patients with PTM [9]. In accordance with our data, they found increased rates of graft loss and decreased overall survival among analyzed US patients with PTM [9]. However, based on the study design and matching strategy, our study further provides more granular data from a German high-volume kidney transplantation center in the Eurotransplant region. Our data adds to existing knowledge by providing evidence of inferior graft survival in PTM-RTx recipients, advocating careful graft surveillance and immunological management in PTM-RTx recipients.

The ever-widening disparity between supply and demand for transplantable grafts has led to ongoing discussions about when PTM patients should be considered eligible for RTx. Analyzing data from the US transplant registry over a 30 year period, Blosser et al. could not find changes in cancer-associated mortality in RTx-recipients despite advances in both immunosuppressive, but also cancer therapy, yet without including patients with a PTM history [12]. Brattström et al. recommend an adaption of waiting times to tumor aggressiveness in RTx-recipients as they identified an increased rate of cancer-associated death in PTM patients, especially in the first five years after diagnosis of cancer [24]. They warrant for waiting times >5 years after diagnosis of cancer as the risk of cancer-associated death in their PTM patients decreased with a longer waiting time [24]. These results were confirmed by Kaufmann et al., concluding that transplantation should rather be delayed in potential RTx-recipients than being performed within the first five years [24,27]. Unterrainer et al., looking at global data from 243 transplant centers, could not find an increased incidence or recurrence of malignancy after different lengths of follow-up [28]. On the other hand, cancer mortality seems to increase in PTM patients, especially during the first years after diagnosis, and there seems to be a link between aggressiveness of PTM and outcome [6,25]. If stratified by aggressiveness, low risk PTM-RTx-recipients had a comparable risk to patients without a history of PTM [6]. Yet, as there are no existing guide lines, Al-Adra et al. published two consensus papers including multiple medical and surgical transplant and oncological experts warranting further databases and following analysis due to recent improvements in both immunosuppressive as well as oncological therapies [5,7]. In the current study, the longest oncological follow-up or waiting time was found in patients with breast cancer, a type of cancer which is considered to be more aggressive than other cancers (e.g., thyroid). Yet, none of these cancers reoccurred, but less aggressive ones such as kidney, non-melanoma skin and urothelial cancer showed recurrence. In addition, patients with a history of breast cancer had longer waiting times as reported by Brattström et al. as well. This can be explained due to the aggressive nature of the tumor and the elevated risk of recurrence [24,29]. Although transplant recipients with PTM experience worse outcomes, increasing current waiting times between successful cancer treatment and transplantation will probably not improve outcomes for these patients. In line with this, Dahle et al. showed similar graft survival with waiting time of only one year and advocated for shorter waiting times in order to overcome increased morbidity and mortality during RRT, even though this might be on the expense of a higher cancer-associated mortality [25]. Yet, RRT bears its own risk for death and associated mortality and some patients may rather decease from RRT-associated comorbidities than cancer-associated death if staying on RRT [25,30].

One advantage for using a case control matched-pair single center data in comparison to large registry data is the ability to provide a more sophisticated analysis on graft specific outcomes by providing granular analysis of associations. The current study provides evidence for inferior one- and five-year overall and death-censored graft survival for PTM-RTx-recipients. This was confirmed by uni- and multivariable analysis, demonstrating a direct correlation between PTM and graft loss. The reason for this remains unknown. One reason might be the acceptance of inferior grafts for PTM-patients. However, data here presented showed no noticeable difference regarding baseline donor characteristics (including KDPI and KDRI) or HLA-mismatches. Another reason for worse graft outcome may be subtle impairment of recipients caused by previous cancer-specific treatment in combination with the impact of RRT and waiting time on the immune system of RTx-recipients [31]. However, waiting times did not differ noticeably between the two cohorts, even though they were slightly longer for PTM-RTx-recipients. Hence, one can exclude that PTM-patients were handled differently on the waiting list, i.e., had longer waiting times, in order to reduce the risk of tumor recurrence. In addition, one might assume that there is potential bias among physicians concerning a restraint against higher immunosuppressive regimes in PTM-RTx patients based on the precaution regarding induction of post-transplant malignancies. However, data on induction and maintenance regimes revealed no differences. It should be noted that our data is in contrast to data reported by Unterrainer et al. who analyzed CTS data and showed a reduced risk for immunological graft losses, which are attributed to be the main cause behind death-censored graft survival in PTM-RTx patients [28]. These results were interpreted as possible evidence for a deficient immunological surveillance capacity in PTM-RTx-recipients, which favored tumor growth before and mitigated homograft rejection after RTx. Unterrainer et al. further identified a tendency towards a better death-censored graft survival in PTM-RTx-recipients. One of the possible explanations for inferior outcomes among prior Tx-recipients with a history of cancer is the use of immunosuppressive therapies, Tx-associated comorbidities, and often less aggressive treatment algorithms because of these comorbidities [32,33].

Most studies that analyze Tx-recipients either compare them to the general population, thereby not taking immunosuppressive therapy and its influence into account, or to patients with HIV or other immunosuppressive diseases [34]. In these cases, however, neither immunosuppressive therapy nor effects of a suppressed immune system in combination with a Tx are taken into account [34]. To address this problem, a retrospective cohort study and matched pair analysis was conducted, thereby excluding the aforementioned biases. Yet, due to its character, this study has to deal with limitations of every retrospective study, potentially resulting in an inclusion bias, especially since the inclusion criteria with PTM were very specific. Due to retrieving data from only one center and the special selection and matching criteria, the cohorts included were quite small and some patients may have been excluded involuntarily. Additionally, despite the fact that the 10-year-inclusion period was not very long, several changes, including changes in procurement procedures and procurement solutions as well as immunosuppressive therapies, took place during this time. As the effect of a PTM was of special interest and should be emphasized especially, we did not compare eligible RTx-candidates or patients remaining on RTT. Consequently, results are only limited in their validity and would need bigger cohorts and a longer inclusion period to be validated. In addition, the relatively small number of 65 patients per group involves an inherent bias when conducting sub-analysis. Thus, while we provide valuable information regarding inferior survival rates in PTM-RTx recipients, is must be stated that the underlying mechanisms for the observed differences remain insufficiently understood and that the included number of patients is insufficient to provide power for in depth sub-analysis. Of course, the comparison with a third cohort remaining on RRT would be interesting. Yet, this study has the strength of including de novo, but also recurrent cancers in the analysis and comparing RTx-recipients not to the general population or non-transplantations-immunosuppressed patients, but to RTx-recipients who were matched for age, sex, time of immunosuppressive therapy or ESRD.

## 5. Conclusions

In conclusion, this matched pair analysis based on retrospective single center data from a German transplant center within the ET region provides first evidence for inferior graft outcomes in RTx-recipients with PTM. While exact causes for an increased graft loss remain unknown, this study advocates focusing on meticulous graft monitoring in PTM patients in addition to a heightened surveillance for cancer recurrence or development of second de novo malignancy.

## Figures and Tables

**Figure 1 jcm-10-02349-f001:**
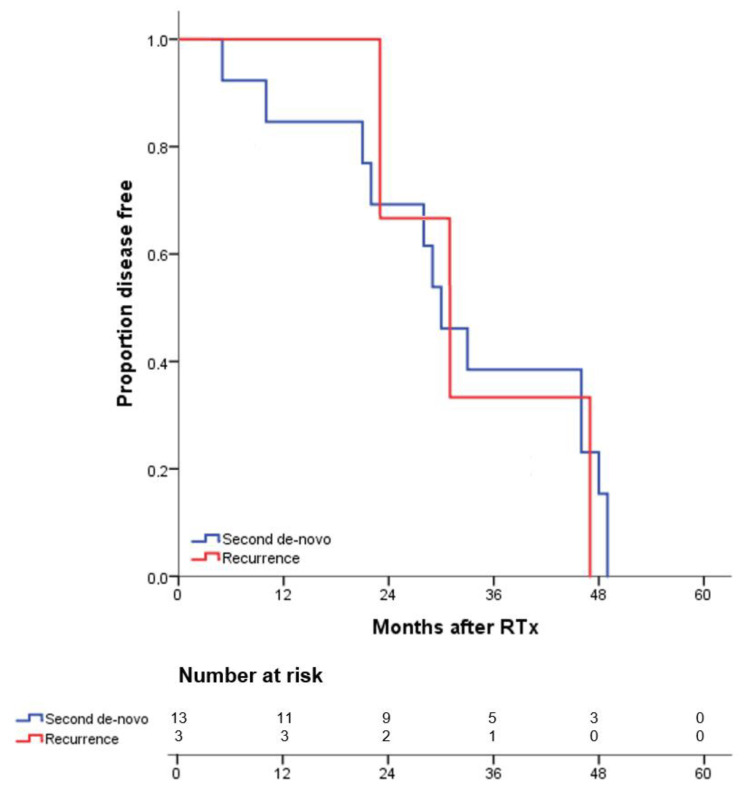
Kaplan-Meier curves for five-year disease-free survival from recurrence and second de-novo malignancy. Kaplan-Meier curves for five-year disease-free survival from recurrence, de-novo or second de-novo malignancy.

**Figure 2 jcm-10-02349-f002:**
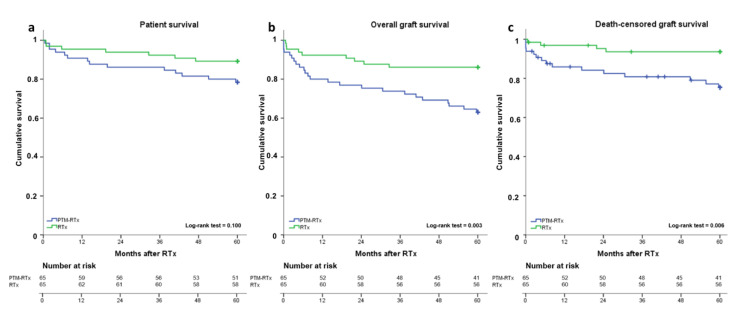
Kaplan-Meier curves for five-year patient and graft survival. Longitudinal patient (**a**), overall graft (**b**) and death-censored graft survival (**c**) stratified for pretransplant malignancy (PTM) renal transplantation (RTx)-recipients and RTx-recipients, respectively. Survival rates of RTx-(green lines) and PTM-RTx-recipients (blue lines) were estimated by Kaplan-Meier methodology and compared by log-rank test.

**Table 1 jcm-10-02349-t001:** Baseline characteristics of recipients with a history of pretransplant malignancy.

PTM-RTx (*n* = 65)		
Age at PTM diagnosis (mean ± SD)		53.5 ± 10.9
Sex (% males)		55.4
Age at RTx (mean ± SD)		62.6 ± 8.6
Time between PTM and RTx (months, mean ± SD (min–max))		105.6 ± 78.1 (6, 468)
Time on RRT (months, mean ± SD)		60.6 ± 31.7
PTM during RRT (*n*, %)		21, 32.3%
PTM before start of RRT (*n*, %)		44, 67.7%
PTM (*n*, %)		Time between PTM and RTx (months, mean ± SD, (min–max))
Skin	13 (20.0)	76.9 ± 62.4 (9, 169)
Urothelial cell	10 (15.4)	133.2 ± 129.9 (27, 468)
Gynecologic	7 (10.8)	178.6 ± 87.6 (34, 301)
Kidney	7 (10.8)	96.7 ± 31.9 (64, 148)
Gastrointestinal	6 (9.2)	100.8 ± 62.9 (50, 206)
Prostate	6 (9.2)	68.7 ± 17.0 (48, 97)
Thyroid	6 (9.2)	63.7 ± 55.0 (6, 155)
Breast	5 (7.7)	136.4 ± 50.6 (101, 225)
Head & Neck	2 (3.1)	109.5 ± 36.1 (84, 135)
Lung	1 (1.5)	78.0
Hematologic	1 (1.5)	129.0
Neuroendocrine	1 (1.5)	33.0

Data are presented as mean ± standard deviation (SD), min, max or relative frequencies. PTM = pretransplant malignancy, RTx = renal transplantation, RRT = renal replacement therapy.

**Table 2 jcm-10-02349-t002:** Oncological outcome for renal transplantation recipients with and without a pretransplant malignancy, matched by age, sex and duration of immunosuppressive therapy.

	PTM-RTx (*n* = 65)	RTx (*n* = 65)	*p*-Value
Post-transplant malignancy (*n*, %)			
De-novo malignancies	-	9 (13.9%)	
Second de-novo malignancies	13 (20)	-	
Recurrence	3 (4.6)	-	0.143 ^b^
No malignancy	49 (75.4)	56 (86.2%)	
RTx to post-transplant malignancy (d, mean ± SD)	973.7 ± 452.4	884.2 ± 496.8	0.285 ^a^
RTx to post-transplant tumor recurrence (d, mean ± SD)	1038 ± 370.9	-	
PNF (*n*, %)	4 (6.2)	2 (3.2)	0.678 ^b^
DGF (*n*, %)	15 (23.1)	12 (18.5)	0.664 ^b^
≥1 BPAR within 1 year after RTx (%)	7 (10.8)	6 (9.2)	1.000 ^b^
1-year eGFR (CKD-EPI, mL/min/1.73 m^2^, mean ± SD)	43.9 ± 19.4	50.5 ± 18.5	0.078 ^a^
5-year eGFR (CKD-EPI, mL/min/1.73 m^2^, mean ± SD)	45.8 ± 19.2	46.5 ±19.2	0.791 ^a^
Graft loss within 1 year after RTx (*n*, %, DC)	9 (13.8)	7 (10.8)	0.688 ^c^
Graft loss within 5 years after RTx (*n*, %, DC)	15 (23.1)	9 (13.8)	0.146 ^c^

Results are presented as mean ± standard deviation (SD) or relative frequencies. Categorical variables were compared using Fisher’s exact test while continuous variables were compared using Student’s *t*-test (normally distributed). PTM = pretransplant malignancy, RTx = renal transplantation, PNF = primary nonfunction, DGF = delayed graft function, BPAR = biopsy proven acute rejection, eGFR = estimated glomerular filtration rate, CKD-EPI = chronic kidney disease epidemiology collaboration, DC = death censored. ^a^ Student’s *t*-test, ^b^ Fisher’s exact test and ^c^ Log-rang test, a *p*-value less than 0.05 was considered statistically noticeable.

**Table 3 jcm-10-02349-t003:** Baseline donor characteristics for renal transplantation recipients with and without a pretransplant malignancy, matched by age, sex and underlying end stage renal disease.

	PTM-RTx (*n* = 65)	RTx (*n* = 65)	*p*-Value
Deceased donor (*n*, %)	58 (89.2)	61 (93.8)	0.508 ^b^
Living donor (*n*, %)	7 (10.8)	4 (6.2)	0.461 ^b^
ABOi (*n*)	2.0	2.0	1.00 ^b^
ESP (*n*, %)	25 (38.5)	28 (43.1)	0.648 ^b^
Donor age (median, IQR)	65 (52.5, 71.5)	64 (52, 70.5)	0.483 ^c^
Donor BMI (median, IQR)	26.2 (24.2, 27.9)	27.6 (29, 42.1)	0.927 ^c^
Donor sex (male, *n*, %)	28 (43.1)	39 (60.0)	0.091 ^b^
Donor creatinine (median, IQR)	1 (0.7, 1.3)	0.9 (0.6, 1.5)	0.772 ^c^
KDRI (mean ± SD)	1.5 ± 0.6	1.4 ± 0.5	0.546 ^a^
KDPI (mean ± SD)	76.8 ± 25.6	72.3 ± 27.5	0.352 ^a^
CIT (h) (mean ± SD)	10 ± 5.0	9.8 ± 4.6	0.572 ^a^
WIT (min) (mean ± SD)	31.8 ± 7.4	33.2 ± 7.0	0.311 ^a^
HLA mismatch (mean ± SD)	3.3 ± 1.6	3.2 ± 1.6	0.746 ^a^
HLA-A mismatch (% 0/1/2)	30.8/53.8/15.4	26.2/52.3/21.5	0.364 ^b^
HLA-B mismatch (% 0/1/2)	13.8/41.5/44.6	20/43.1/36.9	0.249 ^b^
HLA-DR mismatch (% 0/1/2)	23.1/41.5/35.4	23.1/50.8/26.2	0.429 ^b^

Results are presented as mean ± standard deviation (SD), median, interquartile range (IQR) (Q_0.25_–Q_0.75_) or relative frequencies. Categorical variables were compared using Fisher’s exact test while continuous variables were compared using Student’s *t*-test (normally distributed) or Mann-Whitney U test (not normally distributed). PTM = pretransplant malignancy, RTx = renal transplantation, ESP = European Senior Program, AB0i = AB0-incompatible transplantation, BMI = body mass index, KDRI = Kideny Donor Risk Index, KDPI = Kidney Donor Profile Index, CIT = cold ischemia time, WIT = warm ischemia time, HLA = Human Leukocyte Antigen. ^a^ Student’s *t*-test, ^b^ Fisher’s exact test and ^c^ Mann-Whitney U test, a *p*-value less than 0.05 was considered statistically noticeable.

**Table 4 jcm-10-02349-t004:** Baseline recipient characteristics stratified PTM-RTx or RTx, matched by age, sex and underlying end stage renal disease.

	PTM-RTx (*n* = 65)	RTx (*n* = 65)	*p*-Value
Age (mean ± SD)	62.5 ± 8.6	61.9 ± 8.6	0.156 ^a^
Sex (% males)	36 (55.4)	38 (58.5)	0.500 ^b^
BMI (kg/m^2^, mean ± SD)	25.4 ± 3.3	26.9 ± 3.8	0.029 ^a^
RRT (*n*, % yes)	64 (98.5)	64 (98.5)	1.000 ^b^
Dialysis vintage (d, mean ± SD)	1841.9 ± 962.4	2007 ± 1093.4	0.329 ^a^
Hypertension (*n*, %)	59 (90.8)	56 (86.2)	0.508 ^b^
Diabetes (*n*, %)	11 (16.9)	10 (15.4)	1.000 ^b^
CAD (*n*, %)	18 (27.7)	22 (33.8)	0.541 ^b^
PNF (*n*, %)	4 (6.2)	2 (3.1)	0.688 ^b^
DGF (*n*, %)	15 (23.1)	13 (20.0)	0.839 ^b^
≥1 BPAR within 1 year after RTx (%)	7 (10.8)	8 (12.3)	1.000 ^b^
1-year eGFR (CKD-EPI, mL/min/1.73 m^2^, mean ± SD)	43.9 ± 19.4	47.2 ± 16.5	0.286 ^a^
5-year eGFR (CKD-EPI, mL/min/1.73 m^2^, mean ± SD)	45.8 ± 19.2	45.4 ± 16.6	0.446 ^a^
Graft loss within 1 year after RTx (%) DC	9 (13.8%)	2 (3.1)	0.039 ^c^
Graft loss within 5 years after RTx (%) DC	15 (23.1%)	4 (6.2)	0.003 ^c^
Post-transplant malignancy (*n*, %)			
De-novo malignancies	-	11 (16.9)	
Second de-novo malignancies	13 (24.6)	-	
Recurrence	3 (4.6)	-	0.383 ^b^
No malignancy	49 (75.4)	54 (83.1)	
RTx to post-transplant malignancy (days, mean ± SD)	973.7 ± 452.4	1058.9 ± 566.4	0.593 ^a^

Results are presented as mean ± standard deviation (SD) or relative frequencies. Categorical variables were compared using Fisher’s exact test while continuous variables were compared using Student’s *t*-test (normally distributed). PTM = pretransplant malignancy, RTx = renal transplantation, BMI = body mass index, RRT = renal replacement therapy, CAD = coronary artery disease, PNF = primary nonfunction, DGF = delayed graft function, BPAR = biopsy proven rejection, eGFR = estimated glomerula filtration rate, CKD-EPI = chronic kidney disease epidemiology collaboration, DC = death censored. ^a^ Student’s *t*-test, ^b^ Fisher’s exact test and ^c^ Log-rang test, a *p*-value less than 0.05 was considered statistically noticeable.

**Table 5 jcm-10-02349-t005:** Cox proportional hazards regression model with logistic regression analysis of five-year death censored and overall graft survival.

	Death-Censored Graft Survival	Overall Graft Survival
Independent Variables	HR (95% CI) *p*-Value	HR (95% CI) *p*-Value
PTM (yes vs. no)	4.198 (1.392–12.657) 0.011	2.997 (1.393–6.541) 0.005
Recipient age (years)	1.060 (0.995–1.130) 0.072	1.061 (1.011–1.114) 0.016
Recipient sex (male vs. female)	0.878 (0.357–2.162) 0.778	1.078 (0.541–2.151) 0.830
Cold ischemia time (hours)	1.049 (0.963–1.143) 0.273	1.046 (0.980–1.116) 0.176
Warm ischemia time (minutes)	1.037 (0.973–1.106) 0.266	1.059 (1.009–1.111) 0.021
Dialysis vintage (days)	1.000 (0.999–1.000) 0.378	1.000 (0.999–1.000) 0.223
Cause of ESRD	0.752 (0.603–0.937) 0.011	0.846 (0.728–0.983) 0.029
HLA mismatch	1.023 (0.776–1.349) 0.872	1.041 (0.844–1.285) 0.707

HR = hazard ratios, CI = 95% confidence interval. PTM = pre-transplantation malignancy, ESRD = end stage renal disease, HLA = human leukocyte antigen.

## Data Availability

The datasets generated during and/or analyzed during the current study are available from the corresponding author on reasonable request.

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
