# Peer review of "Tumor Recurrence and Graft Survival in Renal Transplant Recipients with a History of Pretransplant Malignancy: A Matched Pair Analysis"

_jcm, 2021, doi:10.3390/jcm10112349_

Round 1
Reviewer 1 Report
The retrospective analysis "Tumor Recurrence and Graft Survival in Renal Transplant Recipients with a History of Pretransplant Malignancy" by Felix Becker presents an overview of a single-center experience with malignancy re- and de-novo occurence after kidney transplantation. The manuscript is well written and the used statistics are adequat. I have two comments.
The authors have done a very deep analysis - however, the number of cases (65 patients per group) is quite low for the amount of performed analyses. It is understandable that the autors aimed to adress as many questions as possible, however it is easy to lose track and the informative value of the subanalyses becomes questionable.
Second, a match analysis can -as mentioned by the authors- also be the basis of bias. Therefor, the overall survival of the unmatched RTx would be of interest. Maybe the authors can mention this in a comment or in the discussion section of the manuscript.
Author Response
Dear Ms. Leng,
Thank you for considering our revised manuscript entitled: Tumor Recurrence and Graft Survival in Renal Transplant Recipients with a History of Pretransplant Malignancy. - A Matched Pair Analysis - (jcm-1206222) for re-submission at Journal of Clinical Medicine.
Firstly, we should like to thank you, and the reviewer for reading through our manuscript and for the excellent and helpful remarks. We appreciate the reviewers’ comments and welcome the opportunity to address these queries and concerns in our revised manuscript. We have addressed all the reviewers’ comments in our responses to both yourself and the reviewer (as per below). We now re-submit to you a revised version of our study, as well as this cover letter, which answers the reviewers’ questions and criticisms, in a point-by-point discussion. The changes made in the revised manuscript are highlighted in yellow for clarity.
We hope that the revised manuscript, together with this point-by-point discussion will now answer all questions and make this revised manuscript suitable for publication in Journal of Clinical Medicine.
Yours Sincerely,
Felix Becker
Reviewer #1
1) The authors have done a very deep analysis - however, the number of cases (65 patients per group) is quite low for the amount of performed analyses. It is understandable that the authors aimed to address as many questions as possible, however it is easy to lose track and the informative value of the subanalyses becomes questionable.
We thank the reviewer for this valuable question and welcome the opportunity to address this important aspect in our revised manuscript.
Accordingly, we have now added a respective statement in our revised discussion to better disclose the fact that extended sub-analyses in such a relatively small cohort is not sufficient to fully address all questions.
Discussion, line 633
Old:
Consequently, results are only limited in their validity and would need bigger cohorts and a longer inclusion period to be validated
New:
Consequently, results are only limited in their validity and would need bigger cohorts and a longer inclusion period to be validated. In addition, the relatively small number of 65 patients per group involves an inherent bias when conducting sub-analysis. Thus, while we provide valuable information regarding inferior survival rates in PTM-RTx recipients, is must be stated that the underlying mechanisms for the observed differences remain insufficiently understood and that the included number of patients is insufficient to provide power for in depth sub- analysis.
2) Second, a match analysis can -as mentioned by the authors- also be the basis of bias. Therefore, the overall survival of the unmatched RTx would be of interest. Maybe the authors can mention this in a comment or in the discussion section of the manuscript.
We agree with the reviewer that performing a matched analysis like ours can be a bias in itself. Moreover, we fully agree with the reviewer that information regarding the overall survival of the unmatched RTx cohort from our center would be of great interest for the readership to better compare survival time of this highly selected cohort with our overall data. In previously published studies from our center, we have analyzed patients who underwent RTx without PTM. Three-year overall survival in these cohorts varied between 88.7% and 92.3%, while 5-year overall-survival varied between 87.1% and 92.4%, respectively [1-3].
References:
- Schutte-Nutgen K, Finke M, Ehlert S, Tholking G, Pavenstadt H, Suwelack B, Palmes D, Bahde R, Koch R, Reuter S (2019) Expanding the donor pool in kidney transplantation: Should organs with acute kidney injury be accepted? -A retrospective study. PLoS One 14 (3):e0213608. doi:10.1371/journal.pone.0213608
- Schutte-Nutgen K, Tholking G, Dahmen M, Becker F, Kebschull L, Schmidt R, Pavenstadt H, Suwelack B, Reuter S (2017) Is there a "weekend effect" in kidney transplantation? PLoS One 12 (12):e0190227. doi:10.1371/journal.pone.0190227
- Mehdorn AS, Reuter S, Suwelack B, Schutte-Nutgen K, Becker F, Senninger N, Palmes D, Vogel T, Bahde R (2020) Comparison of kidney allograft survival in the Eurotransplant senior program after changing the allocation criteria in 2010-A single center experience. PLoS One 15 (7):e0235680. doi:10.1371/journal.pone.0235680
We have now changed the manuscript accordingly and included the following paragraph in our discussion.
Discussion, line 484
Old:
This study provides further evidence for inferior survival rates in PTM-RTx recipients by reporting worse five-year overall patient survival. This confirms previous reports indicating worse outcomes in terms of cancer mortality, all-cause mortality and outcome of posttransplant de novo malignancies in PTM-RTx-recipients [6,9,25].
New:
This study provides further evidence for inferior survival rates in PTM-RTx recipients by reporting worse five-year overall patient survival. Of interest, survival rates in the RTx cohort are comparable to larger cohorts, previously published from our center, in which 5-year overall survival rates between 87.1% and 92.4% were noted. (CITATION) This provides further evidence, that choosing a highly selective cohort for the matched pair analysis did not introduce a bias in the analysis. In general, our survival data confirms previous reports indicating worse outcomes in terms of cancer mortality, all-cause mortality and outcome of posttransplant de novo malignancies in PTM-RTx-recipients [6,9,25].

Reviewer 2 Report
Dear Authors,
I have found this paper interesting and well-presented.
However, I have some remarks/suggestions:
- It should be recognized, also in the abstract, that the reasons underlying the different outcome between PMT-RTx and RTx are not clear
- I wonder that only 75.4% of PMT-RTx received induction therapy? What were the reasons? How was this percentage in the RTx population without a history of cancer? If there is a difference, this factor could have an impact on Tx outcome?
- I think it should be interesting to reconsider the data on clinical outcome excluding the patients with non-melanoma skin cancer
- If I have well understood, there is a mistake in the text (line 269) where the authors declare that “death-censored graft loss within the first year after RTx as well as after 5 years was lower in the 270 PTM-RTx cohort”. Maybe, they refer to death-censored graft survival
- In the Cox analysis, it should be interesting to set as a variable also the time of remission from cancer before RTx
- In figure 1, describing disease-free survival from recurrence or second cancer, I think that there is no sense to include RTxs without a history of previous cancer
Author Response
Dear Ms. Leng,
Thank you for considering our revised manuscript entitled: Tumor Recurrence and Graft Survival in Renal Transplant Recipients with a History of Pretransplant Malignancy. - A Matched Pair Analysis - (jcm-1206222) for re-submission at Journal of Clinical Medicine.
Firstly, we should like to thank you, and the reviewer for reading through our manuscript and for the excellent and helpful remarks. We appreciate the reviewers’ comments and welcome the opportunity to address these queries and concerns in our revised manuscript. We have addressed all the reviewers’ comments in our responses to both yourself and the reviewer (as per below). We now re-submit to you a revised version of our study, as well as this cover letter, which answers the reviewers’ questions and criticisms, in a point-by-point discussion. The changes made in the revised manuscript are highlighted in yellow for clarity.
We hope that the revised manuscript, together with this point-by-point discussion will now answer all questions and make this revised manuscript suitable for publication in Journal of Clinical Medicine.
Yours Sincerely,
Felix Becker
Reviewer #2
1) It should be recognized, also in the abstract, that the reasons underlying the different outcome between PMT-RTx and RTx are not clear.
We appreciate the reviewer’s comment regarding this important aspect, and we fully agree with the reviewer that underlying reasons, responsible for the observed differences between PMT-RTx and RTx are not fully comprehensible. The reviewer correctly alluded to the fact that our granular data does allow to eventually point out correlation and dismiss causation and that this should be better disclosed.
Accordingly, we have now amended the manuscript by adding the following statements:
Abstract, line 46
Old:
This study is the first to show inferior graft survival in PTM-RTx recipients and advocates necessity to focus on more meticulous graft monitoring in PTM recipients in addition to a heightened surveillance for cancer recurrence
New:
Although underlying reasons remain not fully understood, this study is the first to show inferior graft survival in PTM-RTx recipients and advocates for meticulous graft monitoring in PTM recipients in addition to a heightened surveillance for cancer recurrence.
Discussion, line 632
Old:
Consequently, results are only limited in their validity and would need bigger cohorts and a longer inclusion period to be validated
New:
Consequently, results are only limited in their validity and would need bigger cohorts and a longer inclusion period to be validated. In addition, the relatively small number of 65 patients per group involves an inherent bias when conducting sub-analysis. Thus, while we provide valuable information regarding inferior survival rates in PTM-RTx recipients, is must be stated that the underlying mechanisms for the observed differences remain insufficiently understood and that the included number of patients is insufficient to provide power for in depth sub- analysis.
2) I wonder that only 75.4% of PMT-RTx received induction therapy? What were the reasons? How was this percentage in the RTx population without a history of cancer? If there is a difference, this factor could have an impact on Tx outcome?
We thank the reviewer for this excellent question regarding frequencies of induction therapy in PMT-RTx and RTx patients. We now have re-analyzed our data set and have to correct our formerly reported numbers. In fact, frequency of induction therapy in PMT-RTx patients was 83.1% instead of the previously stated 75.4%. We have to apologize for this mistake. At same time, we can assure the reviewer that all other numbers were correct upon an additional checkup.
However, we also found that in the RTx cohort, 54 (83.1%) and 55 (83.3%) patients received an induction therapy, respectively. While this difference was statistically not significant (p = 0.522 vs. PMT-RTx) it remains of interest why overall rate of patients receiving an induction therapy was relatively low among all groups.
Immunosuppressive therapy, consisting of steroids, anti-metabolites, calcineurin and mTOR inhibitors, has been part of the pharmacological strategies to suppress immunreactions since the beginning of transplantation medicine. Antibody induction, however, is a fairly new strategy [4] and it was only routinely introduced at Münster Transplant Centre during the 2000s after the European Commission granted marketing authorisation for basiliximab (Simulect) in October 1998. Thus, the vast majority of patients who received no induction therapy were transplanted during the early 2000s. In addition, many patients were randomized to participate in different trails, evaluating the effect of varies immunosuppressive regimes, of which some omitted induction therapy, i.e. the OSAKA- (Arms 1-3 without induction therapy) [5] or the ADHERE-study (no induction therapy) [6].
References:
- Koyawala N, Silber JH, Rosenbaum PR, Wang W, Hill AS, Reiter JG, Niknam BA, Even-Shoshan O, Bloom RD, Sawinski D, Nazarian S, Trofe-Clark J, Lim MA, Schold JD, Reese PP (2017) Comparing Outcomes between Antibody Induction Therapies in Kidney Transplantation. J Am Soc Nephrol 28 (7):2188-2200. doi:10.1681/ASN.2016070768
- Albano L, Banas B, Klempnauer JL, Glyda M, Viklicky O, Kamar N, Optimising immunoSuppression After Kidney transplantation with ASG (2013) OSAKA trial: a randomized, controlled trial comparing tacrolimus QD and BD in kidney transplantation. Transplantation 96 (10):897-903. doi:10.1097/TP.0b013e3182a203bd
- Rummo OO, Carmellini M, Rostaing L, Oberbauer R, Christiaans MH, Mousson C, Langer RM, Citterio F, Charpentier B, Brown M, Kazeem G, Lehner F, investigators As (2017) ADHERE: randomized controlled trial comparing renal function in de novo kidney transplant recipients receiving prolonged- release tacrolimus plus mycophenolate mofetil or sirolimus. Transpl Int 30 (1):83-95. doi:10.1111/tri.12878
Since we fully agree with the reviewer that this aspect need a more precise description, we have amended the manuscript accordingly by adding the following statement:
Results, line 242
Old:
Induction therapy was given in 49 (75.4%) cases (47 times basiliximab, two times thymoglobuline).
New:
Induction therapy was given in 54 (83.1%) cases (51 times basiliximab, three times thymoglobuline). In the matched cohorts without history of PTM, induction therapy was given to 54 (83.1%) and 55 (83.3) patients, respectively.
Results, line 280
Old
Analyzing the incidence of post-transplant malignancies, immunosuppressive therapy was com-pared and found to be comparable in both cohorts with 95.4% (PTM-RTx) and 100% (RTx) of patients receiving steroids, 93.8% (PTM-RTx) and 89.2% (RTx) MMF, respectively and 80.0% in both cohorts tacrolimus.
New
Analyzing the incidence of post-transplant malignancies, immunosuppressive therapy was compared and found to be comparable in both cohorts with 95.4% (PTM-RTx) and 100% (RTx) of patients receiving steroids, 93.8% (PTM-RTx) and 89.2% (RTx) MMF, respectively and 80.0% in both cohorts tacrolimus. In addition, frequencies of induction therapy were also comparable between PTM-RTx (83.1%) and RTx (83.1%) patients.
Results, line 250
Old:
There was no difference in doses applied and drug levels targets between patients with and without a previous history of malignoma.
New:
There was no difference in doses applied and drug levels targets between patients with and without a previous history of malignoma. In addition, frequency of induction therapy and used drugs was also similar between PMT-RTx and RTx patients.
Discussion, line 472
Old:
In general, era-dependent refinements in induction and maintenance of immunosuppressive therapy could also play a role.
New:
In general, era-dependent refinements in induction and maintenance of immunosuppressive therapy could also play a role. However, both induction therapy as well as maintenance immunosuppressive regimens did not differ between the matched groups, which suggests that differences in tumor recurrence or development of de novo malignancy were rather not attributable to the use of induction therapy.
3) I think it should be interesting to reconsider the data on clinical outcome excluding the patients with non-melanoma skin cancer
We thank the reviewer for this important question. The reviewer correctly alluded to the fact that patients with non-melanoma skin cancer are believed to have a significantly different tumor biology when compared to patients with other skin malignancies and patients with solid or haematological malignancies in general. Therefore, it is a reasonable question whether exclusion of these patients from the cohort of patients with PTM would alter the obtained results.
Accordingly, we have excluded all patients with non-melanoma skin cancer (n=10) from the PTM-RTx cohort (modified (m)PTM-RTx cohort n=55)) and conduced additional analysis to investigate the influence on clinical outcome after omitting these selected patients.
Regarding the primary outcome 5-year patient (mPTM-RTx:76.4 vs PTM-RTx: 78.5%) and death-censored graft (mPTM-RTx: 78.2 vs PTM-RTx: 76.9%) survival, no significant differences were found. Moreover, when analysing secondary outcome parameters, no significant differences were found for frequencies of delayed graft function (mPTM-RTx: 23.6%, vs PTM-RTx: 23.1), frequencies of primary non-function (mPTM-RTx: 5.5%, vs PTM-RTx: 6.2), episodes of biopsy proven acute rejection (mPTM-RTx: 10.9%, vs PTM-RTx: 10.8%) within 1 year after RTx and one (mPTM-RTx: 46.1 ± 26 vs PTM-RTx: 43.9 ± 19.4) and five-year (mPTM-RTx: 48.7 ± 28.3 vs PTM-RTx: 45.8 ± 19.2) eGFR.
In conclusion, excluding patients with non-melanoma skin cancer (n=10) from the PTM-RTx cohort had no influence on clinical outcome.
To share this important aspect with the readership, we have now added the following statement to our manuscript.
Results, line 220
Old:
Skin (ten NMSC, three maligne melanoma) and urothelial cell cancer (n=10) were the two most common types of cancer, followed by gynecological malignancy (n=7) (Table 1).
New:
Skin (ten NMSC, three maligne melanoma) and urothelial cell cancer (n=10) were the two most common types of cancer, followed by gynecological malignancy (n=7) (Table 1). Assuming a different and less aggressive tumor biology in NMSC patients, all analyses were additionally conducted after excluding the ten NMSC patients. However, no significant differences were noted (data not shown) and therefore the patients were included in the final data set.
4) If I have well understood, there is a mistake in the text (line 269) where the authors declare that “death-censored graft loss within the first year after RTx as well as after 5 years was lower in the 270 PTM-RTx cohort”. Maybe, they refer to death-censored graft survival
We thank the reviewer for this remarque and have changed the paragraph accordingly.
Discussion, line 379
Old:
However, death-censored graft loss within the first year after RTx as well as after 5 years was lower in the PTM-RTx cohort (Table 4).
New:
However, death-censored graft survival within the first year after RTx as well as after 5 years was lower in the PTM-RTx cohort (Table 4).
5) In the Cox analysis, it should be interesting to set as a variable also the time of remission from cancer before RTx
We thank the reviewer for this excellent question and welcome the opportunity to improve our manuscript by further exploring this important aspect. Unfortunately, our data set does not include a specific date to define remission. However, we do have the exact date of diagnosis for all PTM-RTx patients. Therefore, we have now conducted further analyses to investigate whether to time from diagnosis to RTx would influence death-censored graft survival or overall graft survival.
We have changed the manuscript accordingly and added the following statement to the manuscript:
Results, line 312
Old:
Since the time-point of diagnosis in relation to RRT as well as waiting time between PTM and RTx are considered to be related to tumor recurrence and patient survival, the PTM-RTx cohort was further stratified accordingly and analyzed for patient and graft survival. The 65 patients in the PTM-RTx cohort were stratified for development of PTM before or during RRT (Supplementary Figure 1) and for a defined waiting time cut off point (>/< 5 years from PTM to RTx) (Supplementary Figure 2).
New:
Since the time-point of diagnosis in relation to RRT as well as waiting time between PTM and RTx are considered to be related to tumor recurrence and patient survival, the PTM-RTx cohort was further stratified accordingly and analyzed for patient and graft survival. The 65 patients in the PTM-RTx cohort were stratified for development of PTM before or during RRT (Supplementary Figure 1) and for a defined waiting time cut off point (>/< 5 years from PTM to RTx) (Supplementary Figure 2). In addition, it was investigated whether time from diagnosis to RTx would influence death-censored graft survival or overall graft survival. Unadjusted Cox proportional hazard modeling revealed no association between time from diagnosis to RTx with death-censored graft survival (HR: 1.002, 95% CI: 0.996 – 1.008, p-value = 0.438)) or overall graft-survival (HR: 1.002, 95% CI: 0.997 – 1.007, p-value: 0.382).
6) In figure 1, describing disease-free survival from recurrence or second cancer, I think that there is no sense to include RTxs without a history of previous cancer
We thank the reviewer for this valuable comment and agree with the reviewer that omitting the RTx cohort without a history of previous cancer is more suitable when describing disease-free survival from recurrence or second cancer.
We have changed figure 1 accordingly.
New figure 1
Changed figure legend, line 343
Old
Figure 1. Kaplan-Meier curves for five-year disease-free survival from recurrence, de-novo or second de-novo malignancy.
New
Figure 1. Kaplan-Meier curves for five-year disease-free survival from recurrence and second de-novo malignancy.
Results, line 302
Old
Kaplan-Meier analysis was used to display disease-free survival for patients with tumor recurrence, de-novo, and second de-novo malignancy (Figure 1). Interestingly, there was only a very slight difference in time between RTx and diagnosis of malignancy between de-novo malignancies in the RTx-cohort (884.2 ± 496.8 days), second de-novo malignancy (973.7 ± 452.4 days) days and tumor recurrence (1038.0 ± 370.9 days) in the PTM-RTx-cohort
New:
Kaplan-Meier analysis was used to display disease-free survival for patients with tumor recurrence and second de-novo malignancy (Figure 1). Interestingly, there was only a very slight difference in time between RTx and diagnosis of malignancy between second de-novo malignancy (973.7 ± 452.4 days) and tumor recurrence (1038.0 ± 370.9 days) in the PTM-RTx-cohort. In addition, time from RTx to diagnosis of de-novo malignancies in the RTx-cohort was also comparable (884.2 ± 496.8 days).
